# MULTI-MODAL DATA SPECTRUM:
# MULTI-MODAL DATASETS ARE MULTI-DIMENSIONAL

**Divyam Madaan**[1]    **Varshan Muhunthan**[1]    **Kyunghyun Cho**[1,2,4]    **Sumit Chopra**[1,2,3]

Courant Institute of Mathematical Sciences, New York University[1]
Center for Data Science, New York University[2]
Grossman School of Medicine, New York University[3]
CIFAR LMB[4]
{divyam.madaan, vm2781, kyunghyun.cho, sumit}@nyu.edu

## ABSTRACT

Understanding the interplay between intra-modality dependencies (the contribution of an individual modality to a target task) and inter-modality dependencies (the relationships between modalities and the target task) is fundamental to advancing multi-modal learning. However, the nature of and interaction between these dependencies within current benchmark evaluations remains poorly characterized. In this work, we present a large-scale empirical study to quantify these dependencies across 23 visual question-answering benchmarks using multi-modal large language models (MLLMs) covering domains such as general and expert knowledge reasoning, optical character recognition, and document understanding. Our findings show that the reliance on vision, question (text), and their interaction varies significantly, both across and within benchmarks. We discover that numerous benchmarks intended to mitigate text-only biases have inadvertently amplified image-only dependencies. This characterization persists across model sizes and types, with models often obtaining high performance by using each modality independently and showing limited dependence on their interaction. We provide a quantitative characterization of multi-modal datasets, enabling a principled approach to multi-modal benchmark design and evaluation.

## 1 INTRODUCTION

Rapid advancement of MLLMs has been accompanied by a significant increase in the number of evaluation benchmarks. A recent survey (Li et al., 2024) identified over 200 multi-modal benchmarks. However, this growth has not been accompanied by a systematic investigation into what these datasets measure. This means the relationships, redundancies, and unique contributions across and within the benchmarks are not well understood. It is often unclear whether a new dataset improves multi-modal evaluation or is largely redundant with existing benchmarks. This ambiguity makes the principled selection of benchmarks for model evaluation a significant challenge.

For example, datasets such as AI2D (Kembhavi et al., 2016), ChartQA (Masry et al., 2022), BLINK (Fu et al., 2024), RealworldQA (xAI, 2024), $V^*$ Bench (Wu and Xie, 2024), TextVQA (Singh et al., 2019) were included in the Gemini 1.5 evaluation (Team et al., 2024), but were omitted from Gemini 2.5 (Comanici et al., 2025) with little justification for the changes. Such inconsistencies in evaluation protocols are common (xAI, 2024; Cohere, 2025), making it difficult to determine whether the reported gains in performance represent true advances in capability or simply adaptation to a different set of benchmark artifacts.

This lack of understanding has led to an inefficient cycle of benchmark development. New datasets are created to address specific uni-modal dependencies (Agrawal et al., 2018), which in turn are found to have new and unforeseen artifacts (Dancette et al., 2021; Si et al., 2022). This process hinders consistent, long-term model comparison and undermines scientific rigor.

Prior work has analyzed the dependence on individual modalities and their interaction in multi-modal models using representation similarity (Kornblith et al., 2019), information-theoretic measures (Tjandrasuwita et al., 2025; Lu, 2023; Madaan et al., 2024), and score-based methods (Gat

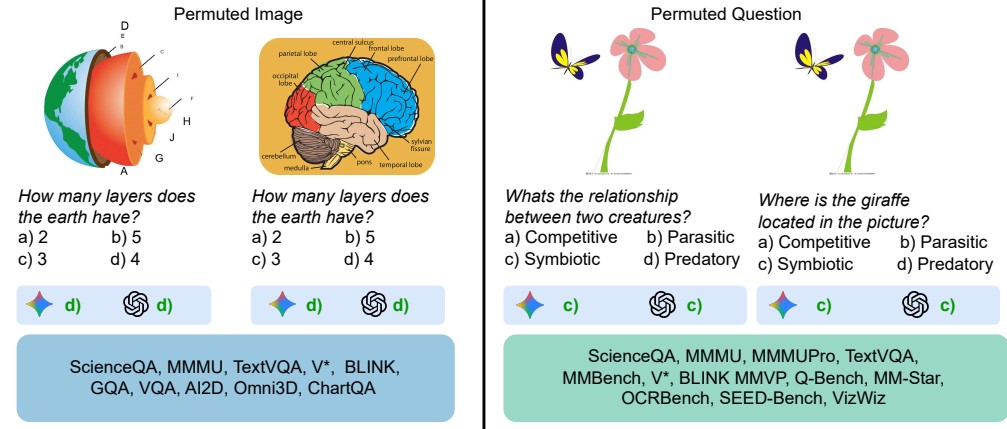

Figure 1: **Demonstration of intra-modality dependencies in multi-modal models using input permutation.** (Left) The models answers about layers of Earth even when the image is replaced by an unrelated diagram of a brain. (Right) The model identifies a symbiotic relationship from the image even when the question is unrelated. These examples highlight a failure of multi-modal reasoning, where models exploit uni-modal priors with the options to obtain an associated answer.

et al., 2021; Parcalabescu and Frank, 2022; Hu et al., 2022; Wenderoth et al., 2025). While providing valuable insights, these studies were limited in scope, focusing on synthetic data, smaller-scale benchmarks such as VQA (Agrawal et al., 2018; Goyal et al., 2017), or earlier generations of models.

To address this gap, we conduct a large-scale empirical study to characterize widely-used multi-modal benchmarks. We hypothesize that these benchmarks evaluate distinct combinations of underlying capabilities. To quantify the multi-dimensional nature of these dependencies, we use intra-modality dependencies (reliance on a single modality for the target task) and inter-modality dependencies (reliance on the interaction between modalities for the target task) based on prior studies (Liang et al., 2023; Madaan et al., 2024). As shown in Figure 1, MLLMs often exploit intra-modality dependencies, answering questions correctly even when a relevant input modality is replaced with corrupted or random data. To quantify modality reliance, we adapt Perceptual Score (Gat et al., 2021). We permute one modality across samples while keeping the other aligned with the labels and measure the resulting performance drop.

Our evaluation spans 23 multiple-choice visual question answering (MCVQA) benchmarks, spanning applications such as general visual question answering, knowledge-based reasoning, real-world spatial understanding, optical character recognition (OCR), and document and chart understanding. We evaluate MLLMs at varying scales, including 8B, 13B, and 34B models (Tong et al., 2024a; Liu et al., 2023; Bai et al., 2025b;a). Our findings confirm our hypothesis; the strength of intra- and inter-modality dependencies vary substantially across and within these benchmarks.

We show that models depend on one input modality while underutilizing the other, rather than using inter-modality dependencies (see Figure 1). We find that many benchmarks designed to mitigate text-only dependencies (Singh et al., 2019; Li et al., 2023a; Tong et al., 2024b; Fu et al., 2024) have inadvertently introduced strong image-only biases, trading one uni-modal shortcut for another rather than evaluating multi-modal reasoning. This issue is not resolved by increasing model scale. On the contrary, larger models often become more adept at using uni-modal dependencies. These results underscore the fundamental limitations of evaluating models with a single aggregate score and highlight the need for a characterization of our evaluation benchmarks based on their strengths of inter- and intra-modality dependencies.

**Contributions.** We conduct the first large-scale empirical analysis of multi-modal dependencies across 23 popular VQA benchmarks. Our analysis shows that these datasets are multi-dimensional regarding their reliance on vision, text, and their interaction, and consequently measure different aspects of multi-modal learning. We find that these differences vary not only across datasets but also within individual benchmarks. To perform this analysis, we use a systematic method to characterize these dependencies. Our results provide a quantitative basis for the design and selection of future multi-modal benchmarks.

## 2 The Multi-modal Spectrum

This section defines inter- and intra- modality dependencies (Section 2.1) for multi-modal learning. We argue that the failure to systematically measure these dependencies has led to an iterative cycle of benchmark design and circumvention (Section 2.2). Existing quantification methods (Section 2.3) do not scale to recent datasets and MLLMs, which motivates our work.

### 2.1 Problem Setup

In supervised multi-modal learning, given a dataset $\mathcal{D} = \{(\mathbf{x}_1^{(i)}, \mathbf{x}_2^{(i)}, \mathbf{y}^{(i)})\}_{i=1}^N$, the goal is to learn a mapping to predict the target label $\mathbf{y}$ from two distinct modalities, $\mathbf{x}_1$ and $\mathbf{x}_2$. The target label $\mathbf{y}$ can be predicted from two distinct dependencies (Liang et al., 2023; Madaan et al., 2024): intra-modality dependency or uniqueness, where $\mathbf{y}$ is dependent on an individual modality, and inter-modality dependency or synergy, where modalities provide joint information not present in isolation. For example, in video-based sentiment analysis, a positive sentiment might be uniquely determined from strong lexical cues within a text transcript alone. In contrast, detecting sarcasm requires interpreting the conflict between the literal semantics of the text and audio or visual expressions of the video.

Following prior work (Liang et al., 2023; Madaan et al., 2024), we model this distinction with a selection variable $\mathbf{v}$ in the multi-modal data generating process, where $\mathbf{v} = 1$ is a mechanism to model the dependencies between the modalities and the target task:

$$p(\mathbf{y}, \mathbf{x}_1, \mathbf{x}_2, \mathbf{v} = 1) = p(\mathbf{y})p(\mathbf{x}_1|\mathbf{y})p(\mathbf{x}_2|\mathbf{y})p(\mathbf{v} = 1|\mathbf{x}_1, \mathbf{x}_2, \mathbf{y}). \tag{1}$$

Although this framework provides a way to separate the effects of individual modalities from their joint combinations, the actual strength of uniqueness and synergy within popular benchmarks and MLLMs remains largely unquantified.

### 2.2 Cat-and-Mouse Game of Benchmark Design

The lack of a principled characterization of these dependencies has resulted in a cat-and-mouse game of benchmark development and subsequent circumvention. This iterative cycle spans a spectrum of multi-modal datasets, from those solvable with a single modality to those that require inter-modality dependencies. To evaluate the multi-modal capabilities of a model, new benchmarks deliberately weaken intra-modality dependencies to necessitate inter-modality dependencies (Goyal et al., 2017; Agrawal et al., 2018; Dancette et al., 2021; Si et al., 2022; Fu et al., 2024; Tong et al., 2024b; Wu and Xie, 2024). Despite these design constraints, models frequently achieve high performance by using intra-modality dependencies. This reliance on intra-modality dependencies is subsequently framed as an exploitation of uni-modal artifacts (Wang et al., 2020; Liang et al., 2023; Zhang et al., 2024b), a behavior that has been assigned labels such as model laziness (Zhang et al., 2024a), modality competition (Huang et al., 2022), or modality greediness (Wu et al., 2022), which prompts further cycles of benchmark revision.

The history of VQA exemplifies this cycle. The original VQA dataset (Antol et al., 2015) contained strong language priors, allowing models to achieve high accuracy by guessing common answers based on the type of questions. To counter this, VQAv2 (Goyal et al., 2017) was introduced, which balanced the dataset by ensuring each question had two images leading to different answers. The subsequent VQA-CP benchmark (Agrawal et al., 2018) further enforced this constraint by changing the answer distribution between the training and test sets to penalize models that relied only on question-based priors. Similarly, the VQA-CE (Dancette et al., 2021) and VQA-VS (Si et al., 2022) datasets were introduced to highlight the prevalence of multi-modal shortcuts in prior VQA benchmarks. This iterative pattern of creation and attack continues with recent benchmarks, such as the progression from MMMU (Yue et al., 2024) to MMMU-Pro (Yue et al., 2025).

Without a systematic way to quantify these dependencies, it is difficult to determine whether the performance of a multi-modal model stems from multi-modal capabilities or from simply exploiting dominant intra-modality dependencies. This ambiguity hinders progress, as we continue to develop complex architectures and algorithms (Li et al., 2021; Wu et al., 2022; Zheng et al., 2023; Liu et al., 2023; Young et al., 2024) without a clear understanding of the spectrum of inter- and intra-modality dependencies in current models and datasets.

### 2.3 QUANTIFYING THE STRENGTH OF DEPENDENCIES

Several quantitative metrics have been developed to measure the dependence of a model on individual modalities. A straightforward approach is to measure performance degradation after shuffling a modality's input at test time, where the resulting performance drop is attributed to that modality's contribution (Gat et al., 2021). More sophisticated methods, such as MM-Shap (Parcalabescu and Frank, 2022), SHAPE (Hu et al., 2022), and InterShap (Wenderoth et al., 2025), use Shapley values to assign importance scores to individual image regions and text tokens, yielding a fine-grained analysis independent of task accuracy.

Despite these advances, no work has systematically positioned recent MLLM evaluation datasets along a continuous multi-modal spectrum defined by their inter- and intra-modality dependencies. In the next section, we adapt a practical methodology based on the perceptual score (Gat et al., 2021) to measure these dependency strengths. We select this method for its simplicity in the two-modality case and its ability to directly compute each modality's marginal contribution. By characterizing datasets along the spectrum of multi-modal dependencies, we can design more targeted benchmarks. Further, we gain deeper insights into model capabilities, paving the way for more robust and generalizable multi-modal systems.

## 3 RECIPE FOR FUTURE DATASETS AND MODELS

Given a multi-modal dataset $\mathcal{D}$ consisting of instances $(\mathbf{x_1}, \mathbf{x_2}, \mathbf{y})$, where $\mathbf{x_1}$ is an image, $\mathbf{x_2}$ is a text, and $\mathbf{y}$ is the ground truth label, we detail a principled evaluation framework inspired by Gat et al. (2021). This requires a baseline multi-modal model $f_\theta$ to evaluate performance, measured by a metric $\mathcal{M}$, under four different input conditions. The chosen baseline model should ideally be a state-of-the-art multi-modal model that has not been trained on the dataset under analysis, thus preventing data leakage.

The four evaluation conditions are:

1. **Paired modalities (Normal):** The model's performance is measured on original, paired data points, $\mathcal{M}(f_\theta(\mathbf{x_1}, \mathbf{x_2}), \mathbf{y})$.
2. **Unimodal (Image only):** The paired text $\mathbf{x_2}$ is replaced with a text instance $\mathbf{x_2'}$ randomly sampled from another data point. Performance on $\mathcal{M}(f_\theta(\mathbf{x_1}, \mathbf{x_2'}), \mathbf{y})$ isolates the informational contribution of the image modality $\mathbf{x_1}$.
3. **Unimodal (Text only):** Symmetrically, the image $\mathbf{x_1}$ is replaced with a random image $\mathbf{x_1}'$. Performance on $\mathcal{M}(f_\theta(\mathbf{x_1'}, \mathbf{x_2}), \mathbf{y})$ isolates the contribution of the text modality $\mathbf{x_2}$.
4. **Both modalities shuffled (Random):** Both modalities are replaced with randomly sampled, uncorrelated instances $(\mathbf{x_1'}, \mathbf{x_2'})$. The model's performance on $\mathcal{M}(f_\theta(\mathbf{x_1'}, \mathbf{x_2'}), \mathbf{y})$ establishes a random baseline.

A dataset that appears balanced at the global level can still contain strong intra-modality dependencies within specific subsets of its data. It is therefore essential that this procedure be supplemented with a more granular analysis of data subgroups. This involves applying the same diagnostic to data subsets categorized by relevant features, such as question type or object categories.

**Rationale for modality shuffling.** We adopt modality shuffling over the option of zeroing out (e.g., using a blank image or an empty string) (Gu et al., 2025) or input perturbation as in prior studies (Hu et al., 2022; Tong et al., 2024a). Zeroing out or adding perturbation creates unnatural, out-of-distribution inputs that elicit unpredictable model behavior, confounding the measurement of inter- and intra-modality dependencies. In contrast, shuffling preserves the marginal distribution of each modality. The model still receives valid inputs, but the inter-modality dependency is broken. The performance metrics derived from this shuffling procedure, visualized in Section 4, enable a direct quantification of inter- and intra-modality dependencies.

**Model-based analysis.** Multi-modal dependencies are a function of both the data and the model interpreting it. Thus, an analysis based on a single model may be confounded by specific inductive biases of that model. To obtain a robust estimate of intrinsic data dependencies, the effect of any single model must be marginalized out. We achieve this using a majority-vote ensemble (Dietterich, 2000) of diverse models to reduce the influence of idiosyncratic model biases.

## 4 EXPERIMENTS

In this section, we describe the evaluation datasets and models used in Section 4.1. Section 4.2 shows the overall performance metrics and Section 4.3 shows the results in various subcategories across multiple datasets.

### 4.1 DATASETS AND MODELS

To assess the capabilities of MLLMs, we use a comprehensive suite of benchmark datasets. Based on the core evaluation skills, we categorize the benchmarks chronologically to show the progression in each category.

- **General visual question answering.** For general VQA, we focus on benchmarks that test real-world and compositional reasoning. We include VizWiz (Gurari et al., 2018), which poses questions from visually impaired users about everyday, uncurated scenes. Following this, we use GQA (Hudson and Manning, 2019) to evaluate visual reasoning and compositional reasoning. To evaluate a wider range of abilities, we incorporate MME (Fu et al., 2023), which covers 14 perception tasks. SEED-Bench (Li et al., 2023a) expands on these with a large-scale multiple choice question format. MMBench (Liu et al., 2024b) further evaluates 20 abilities, including object localization and social reasoning.

- **Expert visual question answering.** To measure performance on tasks requiring specialized knowledge, we evaluate with multiple benchmarks. This includes ScienceQA (Lu et al., 2022), which contains questions from the natural sciences, language and social sciences. We also use MathVista (Lu et al., 2023), which tests mathematical reasoning (logical, arithmetic, and statistical) in diverse visual formats such as word problems, geometric shapes, and plots. For expert-level evaluation, we incorporate MMMU (Yue et al., 2024) and MMMU-Pro (Yue et al., 2025), which consist of college-level problems from exams and textbooks in six core disciplines, probing multi-modal understanding and reasoning.

- **Real-world spatial understanding.** We use Microsoft COCO dataset (Lin et al., 2014) for object recognition and ADE (Zhou et al., 2019) for scene understanding. To measure object-level hallucinations, we use the POPE benchmark (Li et al., 2023b) and measure spatial understanding using RealWorldQA (xAI, 2024). To address the growing importance of temporal reasoning, we include MMVP (Tong et al., 2024b), which tests comprehension and reasoning about long-form video content. Omni3D (Brazil et al., 2023; Tong et al., 2024a) contains the task of determining the depth order and relative distance of 3D objects. Q-Bench (Wu et al., 2024) and BLINK (Fu et al., 2024) evaluate low-level visual perception and general understanding. $V^*$ Bench (Wu and Xie, 2024) focuses on visual grounding in high-resolution images. MM-Star (Chen et al., 2024) is another vision-centric benchmark with human-validated samples to test six fundamental multi-modal capabilities.

- **Optical character recognition (OCR) and document, chart understanding.** We start by evaluating using TextVQA (Singh et al., 2019), which requires models not only to read, but also to reason about text embedded in images. We expand the scope of evaluation with OCRBench (Liu et al., 2024c), which provides a multifaceted assessment that includes text recognition, scene text-centric VQA, document-oriented VQA, key information extraction, and handwritten mathematical expression recognition.

  For document and chart understanding, we evaluate the model's ability to comprehend complex layouts and the relationships between visual elements. We start with AI2D (Kembhavi et al., 2016) for understanding schematic diagrams followed by a ChartQA (Masry et al., 2022), a challenging dataset of human-generated question-answer pairs on various charts and plots.

We use the openly available 8B, 13B, and 34B models from Cambrian-1 (Tong et al., 2024a). These models are built upon Llama-3 8B (Liu et al., 2023), Vicuna-1.5 13B (Chiang et al., 2023), and Nous-Yi 34B (Young et al., 2024) for language processing. For vision, they incorporate a combination of architectures including ViT from SigLIP (Zhai et al., 2023; Radford et al., 2021), DINOv2 (Oquab et al., 2024), and ConvNeXt-XXL (Liu et al., 2022). Our main results are generated by taking a majority vote of the answers among these three models. The code is available online at https://github.com/divyam3897/multimodal-spectrum.

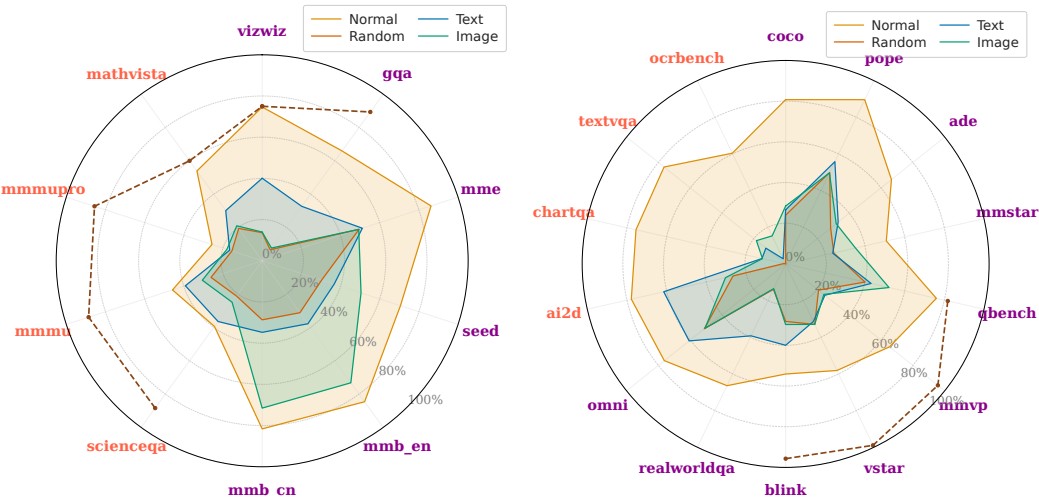

(a) Datasets evaluating visual question answering with **general** and **expert** questions.

(b) Datasets evaluating **spatial understanding** and **OCR, data and chart understanding** .

Figure 2: Radar plot showing the comparison of an ensemble of standard MLLMs with image only, text only and random performance using the recipe from Section 3. The dashed line indicates human performance, which is shown partially due to a lack of data for other benchmarks.

## 4.2 OVERALL RESULTS

Our evaluation in Figure 2 across 23 multi-modal datasets shows that most benchmarks contain both intra- and inter-modality dependencies, allowing models to answer questions without jointly using both the image and the question text. We categorize datasets based on the modality dependencies they contain: (1) inter-modal only, and two non-exclusive categories for datasets that additionally contain (2) text intra-modality dependencies and (3) image intra-modality dependencies.

**Datasets with inter-modality dependency only.** We show that multi-modal datasets with inter-modality dependency only are surprisingly rare. Across all evaluated benchmarks, only four datasets exhibit this characteristic.

*For general and expert question answering, MME (Fu et al., 2023) is the only dataset that demonstrates that permuting one modality makes the task challenging for the model.* For spatial understanding, POPE (Li et al., 2023b), COCO (Lin et al., 2014; Tong et al., 2024a), and V* (Wu and Xie, 2024) datasets contain predominantly inter-modality dependencies. *No datasets in the OCR and chart understanding categories exhibit inter-modality dependencies only.*

The simplest way to curate vision-language inter-modality datasets is to ensure that the answer changes with the change in one modality. This approach has been used in a few binary classification inter-modality datasets (Suhr and Artzi, 2019; Fu et al., 2023; Li et al., 2023b). For instance, POPE and MME contains questions with yes and no answers for the same set of images. This ensures that a model relying on only one modality might correctly answer one question but will fail to correctly answer the corresponding inverse question. This leads to random performance when the inter-modality dependencies are ignored when a modality is shuffled.

**Datasets with text intra-modality dependency.** *Models when evaluated on general and expert knowledge show a reliance on text across all datasets.* For example, models with only the correct input question achieve scores well above random chance on GQA (Hudson and Manning, 2019) (+26%), ScienceQA (Lu et al., 2022) (+17.5%), and MMMU (Yue et al., 2024) (+11.35%), demonstrating that visual input is often not considered necessary by the model for these datasets. This even extends to datasets designed specifically to emphasize visual grounding, such as Blink (Fu et al., 2024), RealWorldQA (xAI, 2024), and Omni3D (Brazil et al., 2023; Tong et al., 2024a). The same pattern holds for OCR, document and chart understanding datasets such as AI2D (Kembhavi et al., 2016), ChartQA (Masry et al., 2022) and TextVQA (Singh et al., 2019), where using question only surpass random performance by 34.94, 11.69 and 12.19 absolute points, respectively.

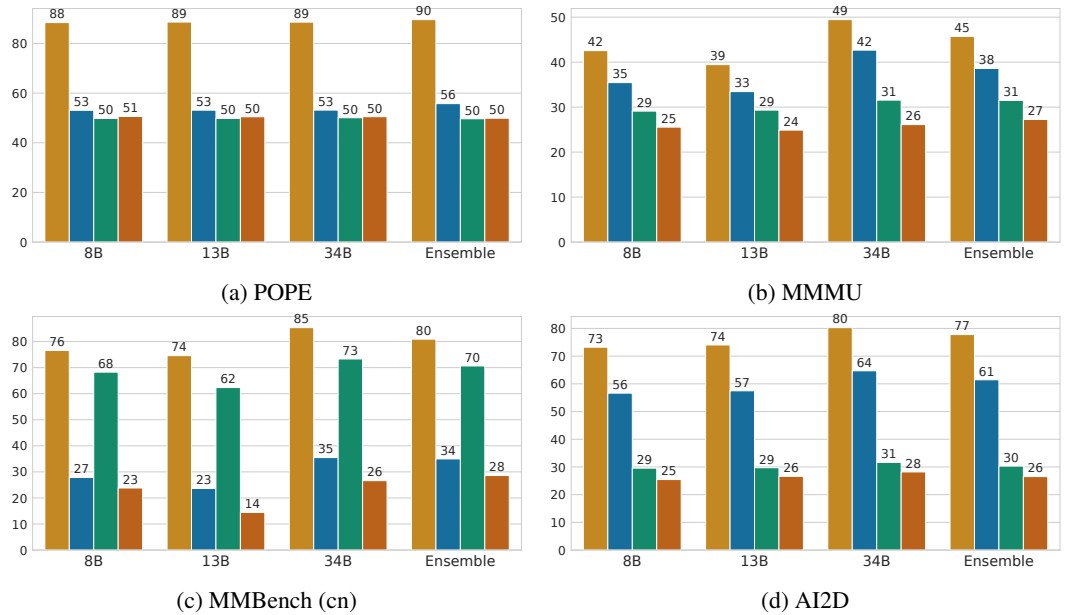

Figure 3: **Effect of model scaling on modality contribution.** Performance of various models (8B, 13B, 34B, and a majority-vote ensemble) on four different datasets. The bars represent standard accuracy and attributed contributions from text, image, and random (bars are in the same order).

These results underscore the challenge of designing benchmarks that do not contain any examples without text-only dependencies. While it is challenging to dissect the precise cause for these biases, many studies have conjectured several issues in data curation. These issues include shortcuts between language and corresponding answers (Goyal et al., 2017), IID train-test splits (Agrawal et al., 2018), shifted prior distributions (Gat et al., 2021), limited human-level perception abilities (Fu et al., 2024) and failures to identify visual patterns in the image (Tong et al., 2024b).

**Datasets with image intra-modality dependency.** *Efforts to eliminate textual biases from benchmarks have led to an unintended consequence of introduction of strong visual intra-modality dependencies.* We find that these newer datasets often allow models to succeed by relying solely on the image, effectively ignoring the question. This is most illustrated in MMBench (Liu et al., 2024b), where an image-only model outperforms a random baseline by 41%. This issue persists even in benchmarks designed to focus on multi-modal reasoning, including TextVQA (Singh et al., 2019), ChartQA (Masry et al., 2022), SEED-Bench (Li et al., 2023a), MMMU-Pro (Yue et al., 2024), MMVP (Tong et al., 2024b), Q-Bench (Wu et al., 2024), and MM-Star (Chen et al., 2024).

Instead of requiring multi-modal understanding, many of these evaluation benchmarks swapped a textual dependency with a visual one to obtain the correct answer. This is because their curation primarily focused on mitigating text-only intra-modality dependencies. We recommend that the central goal of a benchmark design should be to measure the intended task using both modalities for question answering, not to emphasize intra-modality dependencies.

**Effect of model scaling.** Since our analysis is based on a model-dependent accuracy metric, we investigate how modal dependencies change across models of varying scales and architectures. We selected four datasets with distinct dependencies in Figure 3: POPE (Li et al., 2023b), which contains predominantly inter-modality dependencies; MMMU (Yue et al., 2024), which is reliant on both image and text intra-modality dependencies; MMBench (Liu et al., 2024b), dependent on the image modality; and AI2D (Kembhavi et al., 2016), reliant on the text intra-modality dependencies.

We find that uni-modal biases are not consistently mitigated by model scale and can even be exacerbated. For instance, on MMMU (Yue et al., 2024), scaling to a 34B parameter model increased the overall performance and the reliance on both image and text-only dependencies. Similarly, on MMBench (Liu et al., 2024b), larger models exhibit an improved performance with a greater dependency on image-only dependencies (Figure 3). AI2D (Kembhavi et al., 2016) shows a similar trend. The text-only intra-modality dependency increases with scale and persists in the majority-ensemble

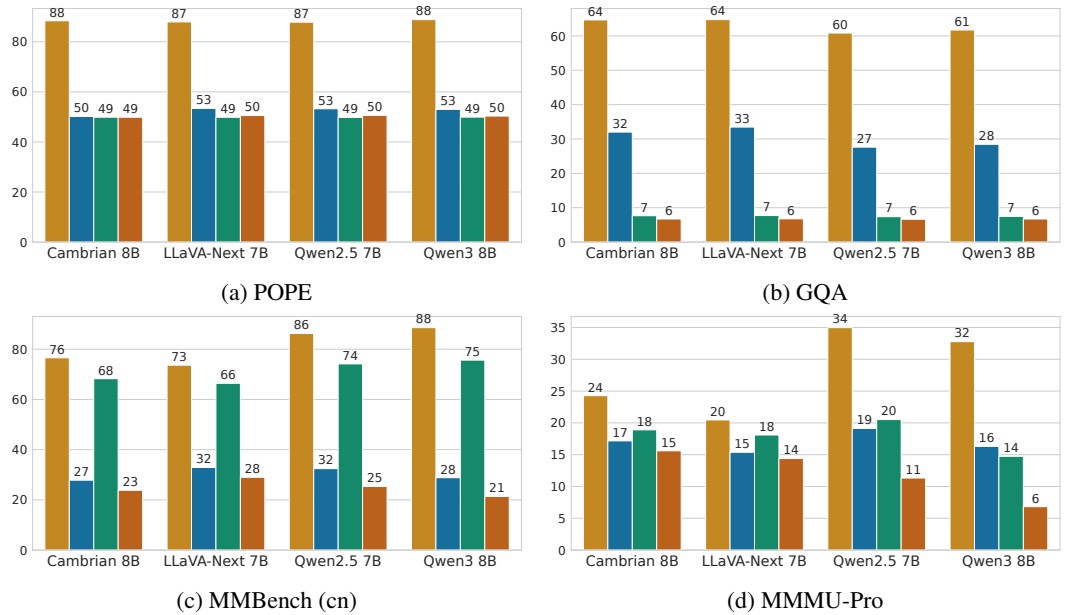

Figure 4: **Effect of model type on modality contribution.** Performance comparison between LLava-Next (May 2024), Cambrian-1 8b (June 2024), Qwen2.5-VL (April 2025) and Qwen3-VL (October 2025) on four datasets selected for their specific dependencies: GQA (text), MMBench (image), POPE (inter-modality) and MMMU-Pro (both image and text). The bars represent standard accuracy and attributed contributions from text, image, and random (bars are in the same order).

model. Conversely, performance on POPE (Li et al., 2023b), a benchmark requiring only inter-modal dependencies, shows no change in performance with increase in model size. We further show results on additional datasets that depend on neither text nor image intra-modality dependencies, as well as datasets that depend on both in Figure B.8. We also include datasets that predominantly depend on image and text intra-modality dependencies in Figure B.9 and Figure B.10 respectively.

**Effect of model types.** We compare four different instruction-tuned models in Figure 4. Particularly, we compare Cambrian-8b released in June 2024 with LLama-3 8B base model (Tong et al., 2024a), LLaVA-Next released in May 2024 with Mistral 7B model (Liu et al., 2024a), Qwen2.5-VL from April 2025 with Qwen2.5 7B language model (Bai et al., 2025b) and Qwen3-VL 8B released recently in November 2025 with Qwen3 language model (Bai et al., 2025a).

Despite substantial differences in the evaluated models and their release dates, we consistently observe intra-modality biases (when present) across all of them. For instance, Qwen models improve the performance on MMBench by around ten percent compared to Cambrian-1 while also increasing the image-only performance significantly. For POPE, as expected, all models exhibit near-random performance when using only image or only text inputs. For GQA and MMMU-Pro, we see comparable levels of biases across different models types. Similar results with additional datasets are shown in Figure B.11.

This findings raise the question of whether improvements in benchmark performance really reflect progress in multi-modal learning or whether models are simply becoming better at using intra-modality dependencies. We hope that our analysis will encourage reporting not only overall benchmark performance but also image-only, text-only, and random baselines, to enable a more holistic evaluation of multi-modal models.

## 4.3 CATEGORY ANALYSIS

Aggregate performance metrics on multi-modal benchmarks can be misleading, often obscuring strong intra-modality dependencies at the sub-category level. As shown in Figure 5, the benchmarks that appeared to use inter-modality dependencies in Figure 2 also contain intra-modality dependencies when evaluated at a granular level.

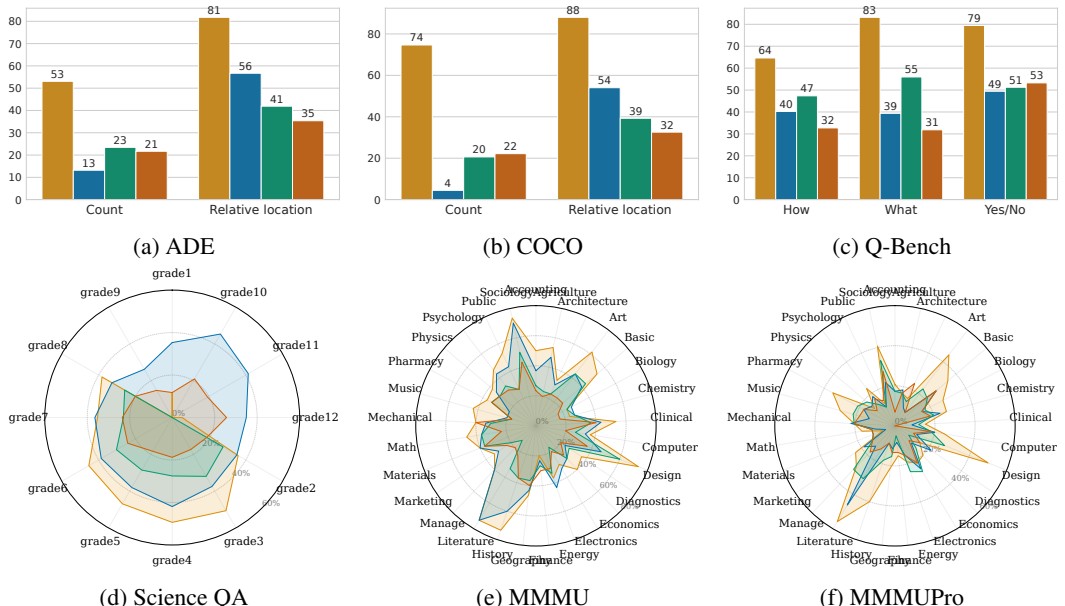

Figure 5: **Analysis of sub-categories across datasets showing dependency on individual modalities.** Although benchmarks may be designed to contain inter-modality dependencies, we show a strong dependence on text intra-modality dependencies for categories such as relative location in ADE and COCO, or higher-grade questions in ScienceQA and multiple categories in MMMU and MMMUPro. This highlights how aggregate metrics can obscure that many instances may not require multi-modal reasoning. We show standard accuracy in yellow and contributions from text in blue, image in green, and random in orange.

This discrepancy is evident across several datasets. In ADE (Zhou et al., 2019) and COCO (Lin et al., 2014; Tong et al., 2024a), while a text-only model's overall performance is only marginally above chance (see Figure 2), it achieves substantial accuracy on the relative location sub-category (Figures 5a and 5b). This phenomenon is amplified in knowledge-intensive benchmarks. In ScienceQA (Lu et al., 2022) (Figure 5d), text-only performance accounts for the majority of the accuracy of questions aimed at grades 10-12. Likewise, many academic subjects within the MMMU and MMMU Pro benchmarks (Yue et al., 2024) (Figures 5e and 5f) contain many instances solvable with a question or an image, respectively, allowing unimodal models to succeed without question or visual information. Conversely, Q-Bench (Wu et al., 2023) (Figure 5c) exhibits the opposite pattern. Individual categories show a dependence on both image and text intra-modality dependencies, yet the aggregate metrics in Figure 2 indicate a notable bias toward the image modality.

These findings are further corroborated by our analysis of datasets such as MME (Fu et al., 2023) and BLINK (Fu et al., 2024) in Figure B.7. We demonstrate that the degree of modality dependence is often inconsistent within a single benchmark. This highlights the multi-dimensional nature of multi-modal datasets, intra- and inter-modality dependencies emerge and vary unpredictably across different sub-populations of the data.

## 5 LIMITATIONS AND FUTURE WORK

Our analysis is constrained by the field's reliance on MCVQA benchmarks. This common practice often fails to test for true multi-modal understanding due to two prevalent failure modes (see Figure 6): text-based intra-modality dependencies, where models ignore the image for factual questions; and image-based intra-modality dependencies, where models select visually correlated answers while disregarding the actual question.

To more holistically evaluate multi-modal capabilities, we propose several crucial future directions. First, we should progress towards building benchmarks that focus on open-ended answer genera-

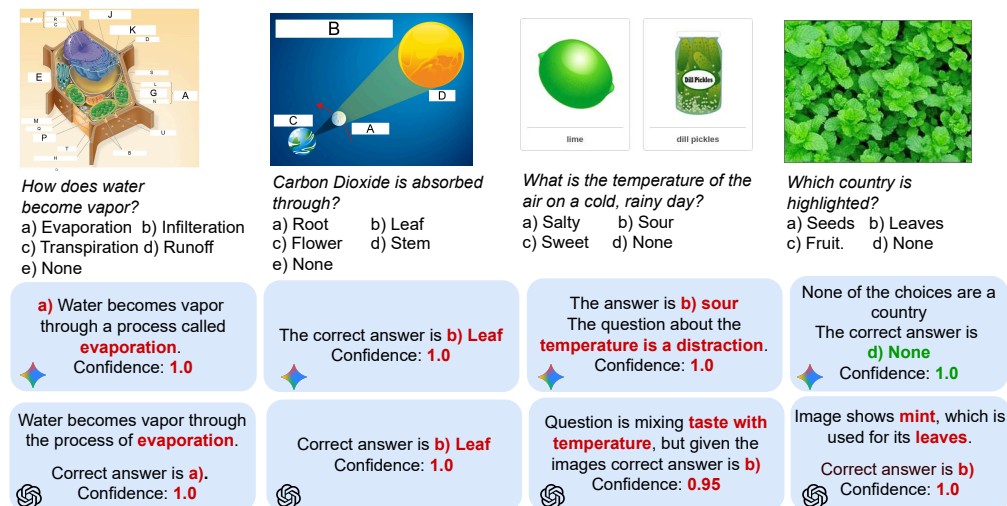

Figure 6: **MLLM failure modes in MCVQA.** Visualization from GPT-5 and Gemini 2.5 Pro showing failure modes in MCVQA, such as relying only on text for factual questions while ignoring the image, or conversely, choosing visually correlated answers while ignoring the question. In all cases, the models were prompted to select one choice and provide a confidence score between 0 and 1.

tion and evaluation (Rei et al., 2020; Balepur et al., 2025). Evaluating free-form responses presents significant challenges. The same meaning can be expressed in many ways, making automated evaluation difficult. This often requires human evaluation, which is slow and expensive. We believe progress in this direction is essential for measuring the necessary multi-modal capabilities.

Second, both benchmarks and models must support the ability to abstain from answering when presented with ambiguous or irrelevant inputs (Whitehead et al., 2022; Feng et al., 2024; Stengel-Eskin et al., 2024). We conduct a preliminary experiment with OpenAI GPT-5 and Google Gemini 2.5 Pro, showing cases where the image or the question was irrelevant to the answer in Figure 6. Despite facilitating abstention by augmenting the instruction set with a "None of the above" option, this approach is largely insufficient to overcome the dependence on uni-modal dependencies for both GPT-5 and Gemini 2.5 Pro models. This highlights that models have a tendency to generate a plausible-sounding but incorrect response over acknowledging ambiguity or lack of information with confidence. Future work should prioritize methods to encourage meaningful abstention.

Lastly, we encourage both future benchmarks and models to report not only aggregated performance but also modality-specific and random baselines to better measure progress. From a benchmark perspective, this helps the community understand how a new dataset compares with existing ones in the inherent biases. From a model perspective, it clarifies where performance gains come from and guides meaningful future improvements.

## 6    CONCLUSION

Our work critically dissects the intra- and inter-modality dependencies of MLLMs on 23 benchmarks. We show that no dataset is truly multi-modal, as each measures multiple dimensions of multi-modal learning with different strengths of intra- and inter-modality dependence. These strengths vary substantially both across benchmarks and across categories within the same benchmark. We find that efforts to mitigate text-based dependencies have often introduced new image-based dependencies, perpetuating a cycle of superficial fixes. This suggests that meaningful progress cannot be achieved simply by developing more benchmarks or chasing leaderboard metrics. Instead, we must critically assess the existing evaluation methods. This includes moving beyond standard multiple-choice formats, incorporating scenarios where models should abstain when they are uncertain, and examining how a model arrives at an answer rather than only what answer it produces.

ACKNOWLEDGEMENT

This work was supported by the Institute of Information & Communications Technology Planning & Evaluation (IITP) with a grant funded by the Ministry of Science and ICT (MSIT) of the Republic of Korea in connection with the Global AI Frontier Lab International Collaborative Research, Samsung Advanced Institute of Technology (under the project Next Generation Deep Learning: From Pattern Recognition to AI), National Science Foundation (NSF) award No. 1922658, Center for Advanced Imaging Innovation and Research (CAI2R), National Center for Biomedical Imaging and Bioengineering operated by NYU Langone Health, and National Institute of Biomedical Imaging and Bioengineering through award number P41EB017183. The computational requirements for this work were supported by NYU IT High Performance Computing resources, services, and staff expertise and NYU Langone High Performance Computing Core's resources and personnel. This work was partly supported in part by the NYUAD Center for Interdisciplinary Data Science & AI (CIDSAI), funded by Tamkeen under the NYUAD Research Institute Award CG016. This content is solely the responsibility of the authors and does not represent the views of the funding agencies.

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

## APPENDIX

**Organization.** In the appendix, we provide the implementation details in Appendix A and additional results in Appendix B.

## A   EXPERIMENTAL DETAILS

**Implementations.** We use the Cambrian-1 (Tong et al., 2024a) open-sourced codebase for all the experiments. We use their publicly released models for evaluation. Datasets like AI2D (Kembhavi et al., 2016), ChartQA (Masry et al., 2022), MMBench (Liu et al., 2024b), MME (Fu et al., 2023), MMMU (Yue et al., 2024), POPE (Li et al., 2023b), RealWorldQA (xAI, 2024), SEED (Li et al., 2023a), TextVQA (Singh et al., 2019), and VizWiz (Gurari et al., 2018) were sourced from LMMS-eval, while others such as ADE, Blink (Fu et al., 2024), COCO (Lin et al., 2014), GQA (Hudson and Manning, 2019), MathVista (Lu et al., 2023), MMMUPro (Yue et al., 2024), MMStar (Chen et al., 2024), MMVP (Tong et al., 2024b), OCRBench (Liu et al., 2024c), Omni3D (Brazil et al., 2023; Tong et al., 2024a), Q-Bench (Wu et al., 2024), ScienceQA (Lu et al., 2022), and $V^*$Bench (Wu and Xie, 2024) were used from their respective sources.

## B   ADDITIONAL RESULTS

The results on different categories for MME (Fu et al., 2023) and BLINK (Fu et al., 2024) datasets are shown in Figure B.7. We further show the effect of model size on additional datasets in Figure B.8, Figure B.9, Figure B.10 and the effect of model types on additional datasets in Figure B.11.

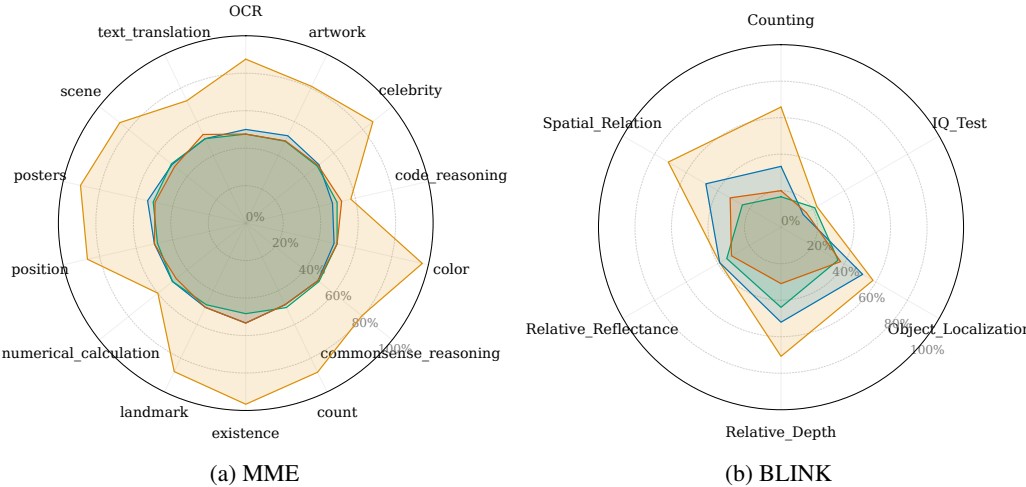

(a) MME                              (b) BLINK

Figure B.7: Analysis of sub-categories for MME and BLINK dataset. We show standard accuracy in yellow and contributions from text in blue, image in green, and random in orange.

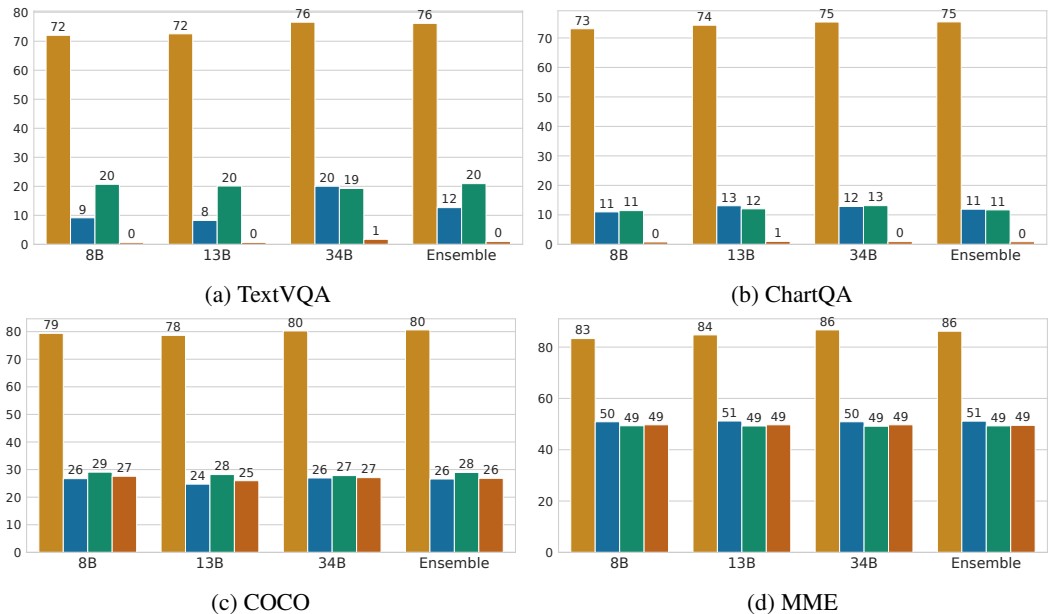

Figure B.8: **Effect of model size on additional datasets.** Performance of various models (8B, 13B, 34B, and a majority-vote ensemble) with both image and text intra-modality dependencies (top) and primarily inter-modality dependency (bottom). The bars represent standard accuracy and contributions from text, image, and random (bars are in the same order).

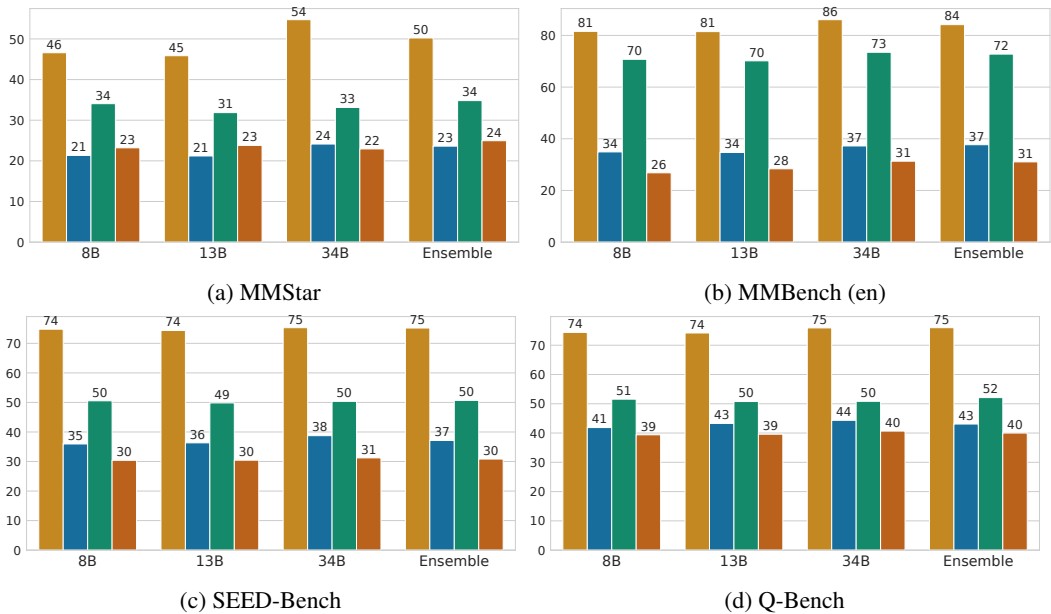

Figure B.9: **Effect of model size on additional datasets with image intra-modality dependency.** Performance of various models (8B, 13B, 34B, and a majority-vote ensemble). The bars represent standard accuracy and contributions from text, image, and random (bars are in the same order).

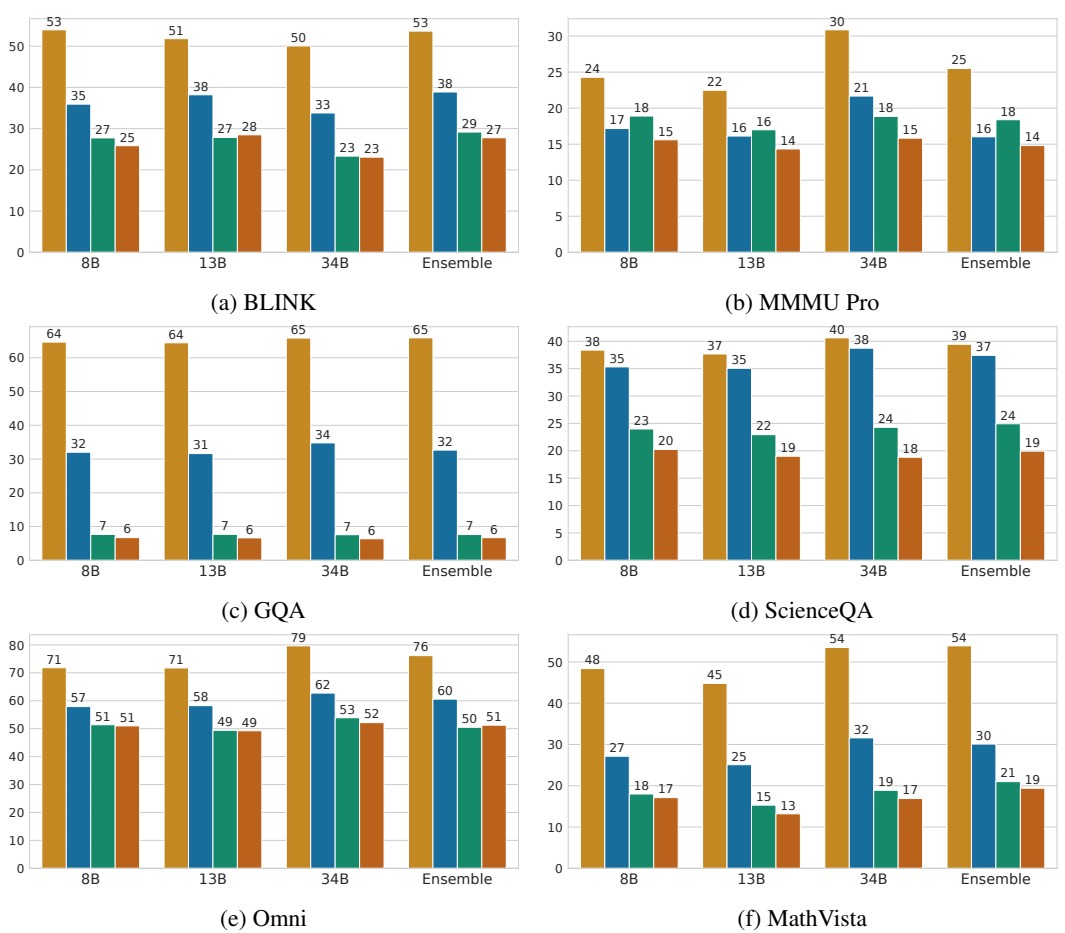

Figure B.10: **Effect of model size on additional datasets with text intra-modality dependency.** Performance of various models (8B, 13B, 34B, and a majority-vote ensemble). The bars represent standard accuracy and contributions from text, image, and random (bars are in the same order).

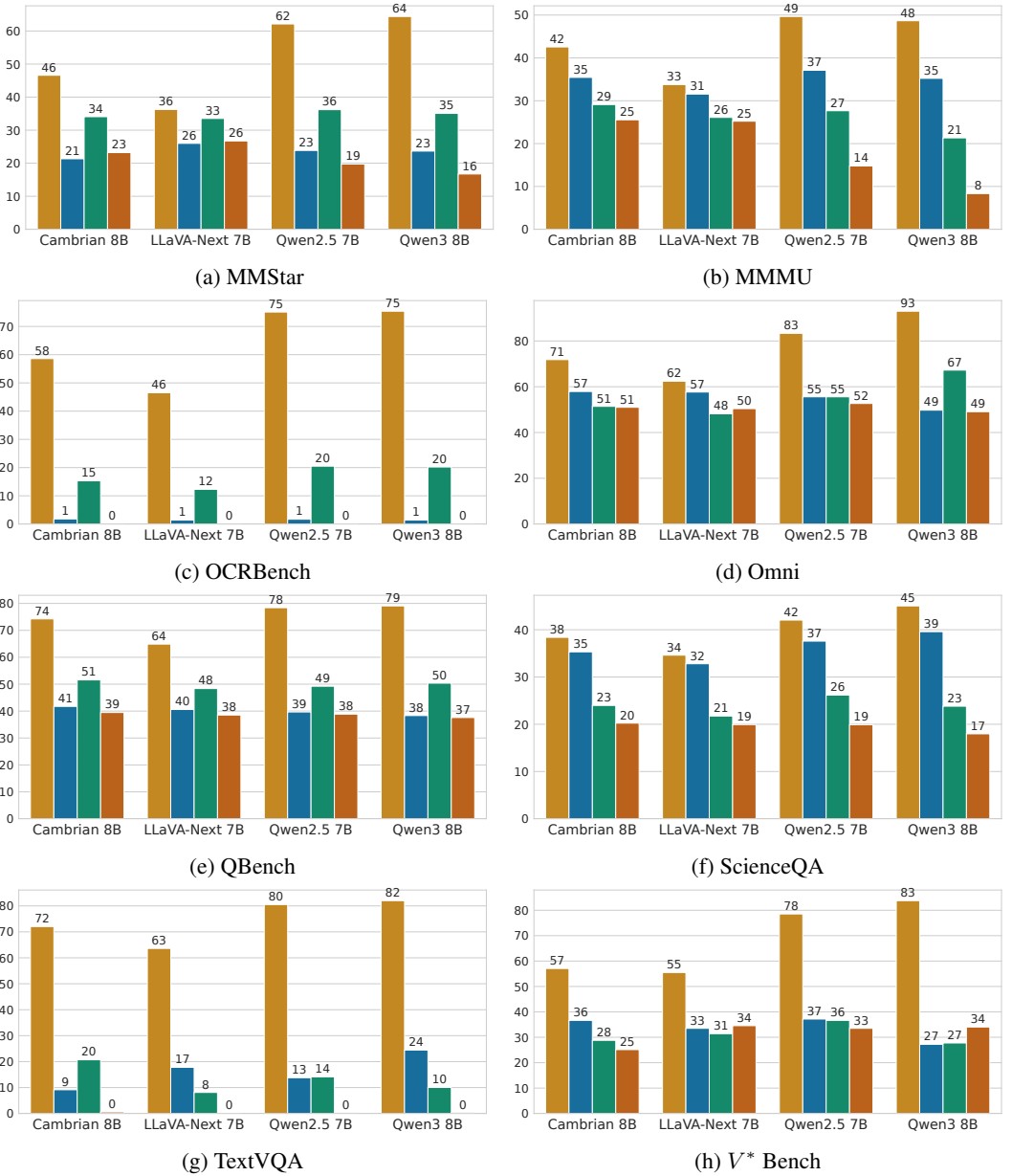

Figure B.11: **Effect of model types on additional datasets.** Performance comparison between various models such as LLava-Next (May 2024), Cambrian-1 8b (June 2024), Qwen2.5-VL (April 2025) and Qwen3-VL (October 2025). The bars represent standard accuracy and attributed contributions from text, image, and random (bars are in the same order).

