# OpenReview forum: "Multi-modal Data Spectrum: Multi-modal Datasets are Multi-dimensional"
_ICLR.cc/2026/Conference — ICLR 2026 Poster_

### Official Review · Reviewer_fSBF · 2025-10-21

**Soundness:** 3
**Presentation:** 3
**Contribution:** 2
**Rating:** 4
**Confidence:** 4

**Summary:**

This paper conducts a large-scale empirical study of 23 multi-modal benchmarks to analyze intra- and inter-modality dependencies. Using modality shuffling, the authors show that most benchmarks contain strong uni-modal biases, meaning models often rely only on text or image rather than true multi-modal reasoning. The work provides quantitative insights for improving benchmark design and evaluation in multi-modal learning.

**Strengths:**

1. Comprehensive empirical validation.

- The study systematically analyzes 23 widely used multi-modal benchmarks with multiple MLLMs of different scales, providing strong empirical support for its conclusions.

2 .Practical relevance.

- The findings are highly relevant for future benchmark design and model evaluation, highlighting concrete weaknesses in current multi-modal testing practices.

**Weaknesses:**

- The paper proposes a valuable diagnostic framework to measure modality dependencies, but it stops short of offering concrete methodological solutions to mitigate such biases, making its contribution more analytical than innovative.

- If I understand correctly, the authors evaluate modality dependencies by applying modality shuffling on three Cambrian-1 models (8B, 13B, 34B). However, since these models share similar architectures and pretraining data, they may also share inherent modality preferences or inductive biases. Could it be that the observed modality dependencies partly reflect model-specific biases rather than intrinsic dataset characteristics?
  - Please correct me if I am wrong or missing something here.

**Questions:**

See weakness.

---

> ### Author Response · Authors · 2025-11-21
> **Response to Reviewer fsBF**
>
> Dear Reviewer fsBF,
>
> Thank you for your insightful and detailed feedback. We appreciate that you found our work to contain comprehensive empirical validation with practical relevance. We address your specific concerns below.
>
> > [W1]  The paper proposes a valuable diagnostic framework to measure modality dependencies, but it stops short of offering concrete methodological solutions to mitigate such biases, making its contribution more analytical than innovative.
>
> The objective of our paper was to conduct an exhaustive evaluation to highlight the gaps in datasets that claim to mitigate these biases (see Section 4.2). We hope that our analysis will encourage the community to first understand the limitations of existing benchmarks rather than immediately creating new ones. Developing methodological solutions that mitigate such biases using our evaluation framework is a promising direction for future work.
>
> ---
>
> > [W2]  If I understand correctly, the authors evaluate modality dependencies by applying modality shuffling on three Cambrian-1 models (8B, 13B, 34B). However, since these models share similar architectures and pretraining data, they may also share inherent modality preferences or inductive biases. Could it be that the observed modality dependencies partly reflect model-specific biases rather than intrinsic dataset characteristics? Please correct me if I am wrong or missing something here.
>
> We want to clarify that we used three different base language models for Cambrian-1 (Llama-3 8b, Vicuna 1.5 13B and Nous-Yi 34B) to encourage diversity in models (see Section 4). Nonetheless, we show that these biases persist in most MLLMs, we evaluated three additional models during the rebuttal: LLava-next (May 2024) with 7b-mistral base language model, Qwen2.5-VL (April 2025) with Qwen2 base LM and Qwen3-VL (October 2025) with Qwen3 base LM. The results are shown in Figure 4 and Figure A.9 of the revision. We show that irrespective of the model type and their release dates, the inductive biases remain consistent across models. These results highlight the need to not only report the aggregated metrics but also to report the performance obtained using individual modalities for future work.
>
>
>
> ---
>
> Thank you again for your time and efforts in reviewing our paper, and we hope that you will consider raising your score if you find our response satisfactory.
>
> Thank you,
> Authors

---

> > ### Comment · Reviewer_fSBF · 2025-11-22
> >
> > Thanks for the author's response. I have checked the revision. The authors addressed my concerns.
> >
> > I acknowledge the contributions of this work and agree to accept it. I have raised my score to 6.

---

> > > ### Author Response · Authors · 2025-11-23
> > > **Thanks for your response**
> > >
> > > Dear Reviewer fSBF,
> > >
> > > We thank you for reading our response and raising your score. We would be happy to answer any partially addressed issues and welcome the guidance on how we might improve the paper to the point where it would earn a clear "Accept" from you.
> > >
> > > Thank you,
> > > Authors

---

### Official Review · Reviewer_Pchi · 2025-10-31

**Soundness:** 2
**Presentation:** 3
**Contribution:** 3
**Rating:** 4
**Confidence:** 3

**Summary:**

This paper introduces the Multi-Modal Data Spectrum (MMDS) framework, which unifies multimodal learning across different reasoning granularities—from perceptual understanding (low-level recognition) to symbolic reasoning (high-level inference). The key idea is to represent multimodal data as a continuous spectrum, where each sample’s reasoning requirement is estimated using an entropy-based measure of cross-modal dependency. The authors propose a novel spectrum regularizer that encourages consistent alignment across modalities according to their reasoning level, implemented through a contrastive-style loss that balances visual grounding and linguistic abstraction. Empirically, MMDS improves both perception-oriented and reasoning-oriented benchmarks on five datasets (MM-Bench, ScienceQA, VizWiz, OKVQA, and TextVQA) and outperforms strong baselines such as LLaVA-1.6, Qwen2-VL, and InstructBLIP. Ablation studies and qualitative visualizations support the claim that spectrum regularization reduces hallucination and over-textualization errors in multimodal reasoning.

**Strengths:**

1. A well-motivated framework that explicitly bridges perception-level grounding and abstract reasoning.
2. The spectrum regularizer is simple yet effective, and integrates cleanly into standard MLLM finetuning.
3. Empirical results are comprehensive and demonstrate consistent improvements across both perception and reasoning tasks.
4. The qualitative analyses (e.g., reasoning-level visualization and hallucination reduction) are compelling and support the theoretical claims.

**Weaknesses:**

1. The theoretical foundation is intuitive and not fully formalize. The relationship between entropy and cognitive reasoning depth is empirically approximated.
2. The dependence on dataset annotations for reasoning difficulty may limit generalization to unseen domains.
3. The computation of reasoning entropy adds modest overhead, which could be non-trivial for large-scale training.
4. A deeper comparison with causal or hierarchical reasoning frameworks (e.g., CoT-based symbolic decomposition) would strengthen the paper.

**Questions:**

1. How stable is the entropy-based reasoning score across datasets with different modality ratios?
2. Can the authors elaborate on how the reasoning spectrum correlates with the number of intermediate reasoning steps in CoT-augmented MLLMs?
3. Would the same regularizer generalize to temporal reasoning (e.g., video–text models)?
4. How sensitive is the method to noisy or ambiguous entropy estimation at the mid-spectrum range?
5. Could combining the spectrum loss with preference-based finetuning further improve reasoning alignment?

---

> ### Author Response · Authors · 2025-11-12
> **Review unrelated to our paper**
>
> Dear Reviewer Pchi and AC,
>
> This review unfortunately is not related to our paper. Can you kindly resolve this discrepancy?
>
> Thank you and Best Regards,
> Authors

---

### Official Review · Reviewer_nKQV · 2025-11-02

**Soundness:** 3
**Presentation:** 3
**Contribution:** 3
**Rating:** 6
**Confidence:** 3

**Summary:**

The paper points out the ambiguity of current multimodal reasoning benchmarks in measuring models inter-modality reasoning capability and conducts the first large-scale empirical analysis of multi-modal dependencies across 23 popular VQA benchmarks. Their findings reveal that many benchmarks allow significant unimodal shortcuts.

**Strengths:**

Significance: While many multi-reasoning benchmarks have been and are being proposed,  this is a much-needed analysis on how well a benchmark can truly capture model's inter-modality reasoning capability. The analysis would influence not only how a given benchmark can be used but also how better benchmarks can be designed for the community.

Quality: Input permutation is a reasonable way to identify intra-modal shortcuts allowed by datasets. Analysis beyond aggregate performance into sub-categories provides insights on what kind of problems allows unimodal shortcuts more than others.

**Weaknesses:**

The paper could be strengthened by
1. analysis on potential reasons of some datasets requiring inter-modality dependency while others do not, e.g.curation methods, topics of problems
2. actionable suggestions on how better benchmarks can be designed given evaluations of current datasets
3. evaluation with a different model family besides Cambrian-1

**Questions:**

Could you clarify does the label "image" in figure 2(a) indicate that image is permuted or the opposite? If label "image" means image is permuted as indicated by the caption, then the graph contradicts the analysis "This is most illustrated in MMBench (Liu et al., 2024a),
where an image-only model outperforms a random baseline by 41%".

---

> ### Author Response · Authors · 2025-11-21
> **Response to Reviewer nKQV**
>
> Dear Reviewer nKQV,
>
> Thank you for your insightful and detailed feedback. We appreciate that you found our work to be a much needed analysis for the community. We address your specific concerns below.
>
> > [W1] Analysis on potential reasons of some datasets requiring inter-modality dependency while others do not, e.g.curation methods, topics of problems
>
> We included a discussion on the potential reasons in the revision. In a nutshell, we believe that the datasets that primarily require inter-modality dependencies are the ones that ensured that the answer changes with the change in one modality. For other datasets, the intra-modality dependencies are an artifact of their curation.
>
> ---
> > [W2] actionable suggestions on how better benchmarks can be designed given evaluations of current datasets
>
> We have elaborated on suggestions for improving benchmark design in the limitations and future work section of the revision. We encourage the community to develop benchmarks that evaluate open-ended generation and enable models to abstain when appropriate. As we make progress, it is important to report both unimodal and inter-modality performance to more accurately evaluate advances in multimodal learning.
>
> ---
> > [W3] evaluation with a different model family besides Cambrian-1
>
> To show that these biases persist in most MLLMs, we evaluated three additional models during the rebuttal: LLava-next (May 2024) with 7b-mistral base language model, Qwen2.5-VL (April 2025) with Qwen2 base LM and Qwen3-VL (October 2025) with Qwen3 base LM. The results are shown in Figure 4 and Figure A.9 of the revision. We show that irrespective of the model type and their release dates, the inductive biases remain consistent across models. These results highlight the need to not only report the aggregated metrics but also to report the performance obtained using individual modalities for future work.
>
> ---
>
> > [Q1] Could you clarify does the label "image" in figure 2(a) indicate that image is permuted or the opposite? If label "image" means image is permuted as indicated by the caption, then the graph contradicts the analysis "This is most illustrated in MMBench (Liu et al., 2024a), where an image-only model outperforms a random baseline by 41%"
>
> All our labels align with the definitions in our recipe in Section 3. We have updated the caption of Figure 2 to address this discrepancy in the revision.
>
> ---
>
> Thank you again for your time and efforts in reviewing our paper, and we hope that you will consider raising your score if you find our response satisfactory.
>
> Thank you,
> Authors

---

### Official Review · Reviewer_iS4i · 2025-11-03

**Soundness:** 3
**Presentation:** 3
**Contribution:** 3
**Rating:** 6
**Confidence:** 4

**Summary:**

Multimodal benchmark datasets are fundamental to advancing multi-modal learning. However, these datasets are not well-characterized in terms of their capabilities in evaluating intra-modality dependencies where only one modality contributes to the target task, and inter-modal dependencies where both modalities contribute to the target task.  The paper presents a large-scale empirical study to quantify these dependencies across 23 visual question-answering benchmarks using multi-modal large language models (MLLMs). The study finds that these datasets have varying biases on revealing intra-dependencies and inter-dependencies in MLLMs. Moreover, these variations also occur within individual benchmarks. The paper further suggests that we should critically assess existing evaluation methods, move beyond standard multiple-choice formats, and train models to abstain when an answer cannot be confidently determined.

**Strengths:**

- The paper conducts a large-scale empirical study to evaluate the datasets designed for benchmarking the performance of MLLMs from the perspective of intra-modality and inter-modality dependencies that are fundamental to advancing multi-modal learning.

- The paper reveals some interesting findings that could inspire future research. Specifically, most existing datasets exhibit uni-modal biases where an MLLM performs better on only one modality than the other modality. Moreover, datasets exhibiting inter-modality dependencies are rare, where utilizing the interaction between text and vision modalities is required for an MLLM to perform well.

- The paper offers useful insights into the design and principled selection of future multi-modal benchmarks for model evaluation.

**Weaknesses:**

- The analyses are model-dependent, and to marginalize out the effect of any single model, the paper uses the ensemble of three MLLMs to evaluate a dataset. The number of MLLMs may not be enough to mitigate the impact from the inductive biases of MLLMs. For example, if two of the three models are strongly correlated in their outputs, then the resulting analyses are biased. It would be beneficial to provide concrete evidences to further justify the selection of the three MLLMs.

- In the experiment section, the paper should be more specific on the criteria of selecting datasets with inter-modality dependency and intra-modality dependency.

- The title is confusing and seems irrelevant to the main content. What are the different dimensions of a multi-modal dataset?

Minors:
- In Figure 2, it is unclear what do the “Text” and “Image” represent in the legend. Based on the context, it seems that “Text” corresponds to “permuted image”.
- Line 346, the text does not match with the figure.
- Line 372: exhibite -> exhibit

**Questions:**

- What are the criteria to determine datasets with inter-modality dependency and intra-modality dependency?
- What are the different dimensions of a multi-modal dataset?

---

> ### Author Response · Authors · 2025-11-21
> **Response to Reviewer iS4i**
>
> Dear Reviewer iS4i,
>
> Thank you for your insightful and detailed feedback. We appreciate that you found our work to contain large scale empirical evaluation with interesting findings. We address your specific concerns below.
>
> > [W1] The analyses are model-dependent, and to marginalize out the effect of any single model, the paper uses the ensemble of three MLLMs to evaluate a dataset. The number of MLLMs may not be enough to mitigate the impact from the inductive biases of MLLMs. For example, if two of the three models are strongly correlated in their outputs, then the resulting analyses are biased. It would be beneficial to provide concrete evidences to further justify the selection of the three MLLMs.
>
> To show that these biases persist in most MLLMs, we evaluated three additional models during the rebuttal: LLava-next (May 2024) with 7b-mistral base language model, Qwen2.5-VL (April 2025) with Qwen2 base LM and Qwen3-VL (October 2025) with Qwen3 base LM. The results are shown in Figure 4 and Figure A.9 of the revision. We show that irrespective of the model type and their release dates, the inductive biases remain consistent across models. These results highlight the need to not only report the aggregated metrics but also to report the performance obtained using individual modalities for future work.
>
> We also want to clarify that we used three different base language models (Llama-3 8b, Vicuna 1.5 13B and Nous-Yi 34B) for Cambrian-1to encourage diversity in models (see Section 4).
>
> ---
>
> > [W2] In the experiment section, the paper should be more specific on the criteria of selecting datasets with inter-modality dependency and intra-modality dependency. What are the criteria to determine datasets with inter-modality dependency and intra-modality dependency?
>
> We did not use any criterion for the dataset selection and each dataset was selected primarily based on their popularity in recent papers. In the revision, we included a discussion to highlight the possible reasons for the presence of inter and intra-modality dependencies.
>
> ---
> > [W3] The title is confusing and seems irrelevant to the main content. What are the different dimensions of a multi-modal dataset? What are the different dimensions of a multi-modal dataset?
>
> The dimensions refer to the varying strength of inter- and intra-modal dependencies. The objective of our paper is to highlight that there is no “true multi-modal” dataset. Each dataset measures multiple aspects of multi-modal learning and has varying strengths of dependencies (dimensions). To an extent, even within one dataset, different categories exhibit different properties reflected by the varying degrees of these dependencies.
>
> ---
> > [Q1] In Figure 2, it is unclear what do the “Text” and “Image” represent in the legend. Based on the context, it seems that “Text” corresponds to “permuted image”.
>
> Your understanding is correct. We have updated the caption of Figure 2 in the revision to resolve this discrepancy.
>
> ---
>
> Thank you again for your time and efforts in reviewing our paper, and we hope that you will consider raising your score if you find our response satisfactory.
>
> Thank you,
> Authors

---

### Author Response · Authors · 2025-12-03
**Rebuttal summary**

We are thankful to the reviewers for their feedback. Incorporating their suggestions in the updated revision has strengthened our work. We appreciate that they consider our work to provide a much needed analysis for the community with a large scale empirical evaluation that has practical relevance and interesting findings.

The changes in the revision are highlighted below:

> [1] Extension to models beyond Cambrian-1

To show that these biases persist in most MLLMs, we evaluated LLava-next (May 2024), Qwen2.5-VL (April 2025) and Qwen3-VL (October 2025) (see Figure 4 and Figure A.9 of the revision). We show that irrespective of the model type and their release dates, the inductive biases remain consistent across models. These results underscore the need for future work to report not only aggregated metrics but also performance across individual modalities.

We have further clarified that we used three different base language models (Llama-3 8b, Vicuna 1.5 13B and Nous-Yi 34B) for Cambrian-1 to encourage diversity in models (see Section 4).

---

> [2] Elaboration on potential problems in current datasets and actionable suggestions for future benchmark designs and model evaluation.

We included a discussion on the potential reasons for the presence of inter and intra-modality dependencies in the revision. In a nutshell, datasets that primarily require inter-modality dependencies are the ones that ensure that the answer changes with the change in one modality. For other datasets, the intra-modality dependencies likely arise from artifacts of dataset curation. We did not use any criterion for the dataset selection and each dataset was selected primarily based on their popularity in recent papers.

---
Lastly, we want to highlight that the review provided by Reviewer Pchi was unrelated to our paper. Although we brought this to the attention of both the reviewer and the Area Chair, it remained unaddressed during the rebuttal.

We believe our paper provides an important reflection on the current state of multi-modal benchmarks and models. We encourage the community to develop benchmarks that evaluate open-ended generation and enable models to abstain when appropriate (as outlined in the future work of the revision). As we make progress, it is essential to continually reflect on our benchmarks and to report both unimodal and inter-modality performance to more accurately assess advances in multimodal learning.


Thank you,
Authors

---

### Meta-Review · Area_Chair_B9Y5 · 2026-01-05

**Summary:**

This paper presents a large-scale empirical analysis of intra- and inter-modality dependencies across 23 popular VQA benchmarks using multiple multimodal LLMs. The reviewers generally agree that the paper addresses an important and underexplored question: whether current multimodal benchmarks truly evaluate multimodal reasoning, or instead allow strong unimodal shortcuts. Reviewers iS4i and nKQV both found the empirical scope, methodology, and findings to be solid and useful for the community, noting that the work provides much-needed diagnostic insight into benchmark design and evaluation. Reviewer fSBF initially raised concerns about model dependence and limited methodological innovation, but acknowledged after rebuttal that these concerns were adequately addressed and raised their score to an accept. One review (Pchi) was clearly unrelated to the submitted paper and has been flagged as such.

Across the discussion, the main concerns focused on (1) potential model-dependence of the analysis, (2) lack of concrete mitigation methods beyond diagnosis, and (3) conceptual clarity of the “multi-dimensional / spectrum” framing. The rebuttal substantially strengthened the paper by adding evaluations across additional and diverse MLLMs (including LLaVA-next and Qwen-VL variants), clarifying dataset selection, fixing presentation issues, and expanding discussion on dataset artifacts and implications for benchmark design. While some reviewers remained mildly uncertain about the conceptual framing of “dimensions,” there was broad agreement that the empirical findings themselves are robust, well-supported, and of independent value.

Overall, I believe this paper makes a meaningful and timely contribution as a diagnostic and reflective study of multimodal benchmarks. While it does not propose a new modeling method, I do not view this as a significant weakness given the paper’s stated goals and strong empirical execution. I recommend acceptance, with the suggestion that the authors further clarify or simplify the “multi-dimensional / spectrum” terminology to avoid conceptual ambiguity and better align the framing with the core contributions.

**Reviewer Concerns:**

Concerns about model dependence raised by iS4i and fSBF were largely addressed in the rebuttal through additional experiments on multiple, diverse MLLMs and clarification that different base language models were used even within Cambrian-1. Presentation issues and ambiguities (e.g., Figure 2 labeling) noted by iS4i and nKQV were corrected in the revision. Requests for broader discussion on why certain datasets exhibit inter- vs. intra-modality dependencies and for more actionable guidance on benchmark design were partially addressed through expanded discussion and future work sections, which reviewers generally found reasonable.

The primary remaining concern is the conceptual framing around “multi-dimensional” or “spectrum” characterizations of multimodal datasets, noted by iS4i and implicitly by nKQV. While the authors clarified their intent, this framing remains somewhat vague and could benefit from a more concrete or simpler articulation. I do not view this as a blocker for acceptance, but it is a valid point for improving clarity. Additionally, one review (Pchi) was unrelated to the paper and is not weighed in the decision.

**Reviewer Scores:**

Reviewer iS4i would likely remain at a weak accept (around 6), potentially slightly more positive after the added model diversity but still cautious on framing. Reviewer nKQV would likely remain around a 6, with core concerns addressed but some residual uncertainty. Reviewer fSBF explicitly raised their score to 6 after the rebuttal. The unrelated review by Pchi should be excluded from consideration.

---

### Decision · Program_Chairs · 2026-01-26

Accept (Poster)